# Carbon Nanotube (CNT) Encapsulated Magnesium-Based Nanocomposites to Improve Mechanical, Degradation and Antibacterial Performances for Biomedical Device Applications

**Jinguo Zhao [1], Ma Haowei [2,*], Abbas Saberi [3,*], Zahra Heydari [4] and Madalina Simona Baltatu [5,*]**

1 School of Mechanical Engineering, Xijing University, Xi'an 710123, China
2 Department of Mechanical Engineering, Faculty of Engineering, University of Malaya, Kuala Lumpur 50603, Malaysia
3 Department of Materials Engineering, South Tehran Branch, Islamic Azad University, Tehran 1777613651, Iran
4 School of Electrical and Computer Engineering, College of Engineering, University of Tehran, Tehran 1439957131, Iran
5 Department of Technologies and Equipment for Materials Processing, Faculty of Materials Science and Engineering, Gheorghe Asachi Technical University of Iaşi, Blvd. Mangeron, No. 51, 700050 Iasi, Romania
* Correspondence: mhwfangzhang@gmail.com (M.H.); abbassaberi65@gmail.com (A.S.); madalina-simona.baltatu@academic.tuiasi.ro (M.S.B.)

**Abstract:** Nowadays, magnesium (Mg) composites are gaining much attention in biomedical device applications due to their biocompatibility and biodegradability properties. This research is to study the microstructure, mechanical, corrosive and antibacterial properties of Mg−2.5Zn−0.5Zr/xCNT (x = 0, 0.3, 0.6, 0.9) composites made with mechanical alloying and semi-powder metallurgy (SPM) processes, accompanied by SPS. Based on the microstructural characteristics, CNTs were almost uniformly distributed in the Mg matrix. The results displayed that the hardness and ultimate compressive strength (UCS) of the composites were meaningfully increased compared to a Mg matrix. Moreover, the degradation rate of Mg composites was almost halved in the presence of small amounts of CNTs in the Kokubo simulated body fluid (SBF). Due to the slowed degradation process, the Mg−2.5Zn−0.5Zr/0.6CNT biocomposites exhibited excellent cellular compatibility. Evaluation of antibacterial activity displayed that adding CNTs to the Mg matrix could significantly prevent the growing of Escherichia coli (E. coli) and Staphylococcus aureus (S. aureus). In general, the research results showed that CNTs are an efficient reinforcement for Mg−2.5Zn−0.5Zr/CNTs biocomposites, which leads to improved mechanical, degradation and antibacterial performances.

**Keywords:** nanocomposites; magnesium; CNTs; mechanical properties; biocompatibility

## 1. Introduction

Over the past decades, researchers have shown an increasing interest in replacing conventional implantable devices with Mg-based alloys and composites to overcome common problems in orthopedics [1–3]. Mg, on the other hand, is found naturally in bone tissue and is essential for the body's metabolism [4]. Mg is the fourth most plentiful ion in the human body, so a person weighing 70 kg needs about 1 mole of Mg, half of which is stored in bone tissue.

It is also possible to mention the elastic modulus of Mg (41–45 GPa) compared to iron (Fe) (211.4 GPa) or zinc (Zn) (90 GPa), which makes it further analogous to the natural bone of the body (3–20 GPa) and prevents stress shielding phenomena [3,5]. Mg is well involved in the reaction of bone minerals and controls bone reproduction and regeneration [6]. Mg, on the other hand, is easily destroyed in the body and excreted through the body fluids, consequently there is no need for a second surgery to remove it from the body [7,8]. Of course, in the biological environment of the body, the decomposition rate of Mg with a decrease in mechanical properties is higher than the recovery rate of bone tissue, while

the release of hydrogen gas is observed also on the surface of Mg. Furthermore, the release of hydrogen gas on the surface of Mg is also one of the things that limits the use of Mg in complex physiological environments [8–10]. Moreover, Mg-based composites are less resistant to bacteria, causing infections around implantable devices and triggering more problems for postoperative patients. Bone infections are fundamental and inevitable problems that put people under the stress of medical and financial difficulties. It is generally triggered via a prime microbial infection, mostly by S. aureus, and is really an inflammatory process leading to bone demolition [11,12].

Therefore, the mechanical properties, degradation and antibacterial performances of Mg samples must be improved in some way. Alloying and composite production at the same time can be a suitable method to improve the properties of Mg [5,13]. When choosing alloying elements as background, it should be noted that the elements ought not to be toxic in the body [14,15].

Zn is an essential element for the survival of human life. Zn is one of the foremost vital elements in the body after Fe. This element is stored in muscle and is also present in blood cells, retina, bone, skin, kidney and liver [16]. Zn is also an indispensable element for cell growth and an important component of bones in the body [16]. Studies have shown that Zn increases osteoblast adhesion, cell proliferation and differentiation in bone cells. Studies have also shown that adding Zn to Mg in the range of 1–5 wt% improves the mechanical properties and corrosion resistance of the alloy [16,17]. On the other hand, Zn as an experimental implantable material poses no risk of toxicity to humans. Zn poisoning to humans has only been observed with overdose (50 mg per day) or excessive toxic exposure to Zn, whereas whole destruction of pure Zn implants releases only 150 μg/day [18]. Zirconium (Zr) is another biocompatible alloying element with a grain refinement function. Recently, Mg-Zr alloys have received much attention for their great specific damping capacity (80%), which helps to dampen the vibrations created at the implant/bone interface during movement and loading [19,20]. Gu et al. [21] has shown that adding 1 wt% of Zr in Mg has led to substantial progress in the strength and ductility of the metal (UTS = 171.87 ± 2.31 Mpa), an increase of 27 ± 2% in elongation and a reduction of 50% in the degradation rate. Generally, the alloying content of Zr ought to be less than 1 wt% in Mg-based biomedical alloys [4]. Based upon these considerations, a basic chemical composition containing Zn and Zr is proposed in this study for Mg alloys.

As previously mentioned, one of the most popular techniques to achieve the superior properties of Mg is the encapsulation and fabrication of composites using suitable nanofillers [13]. It has been determined that due to the attractive properties of CNTs, including Young's modulus (1TPa), extremely high strength (30 GPa) [22], stiffness (1TPa) in tension [22], load transfer efficiency, chemical inertness and high thermal conductivity (3000 W/m.K) [23], and large surface-to-volume ratio, these are a very high potential candidates as a reinforcing material for multipurpose composites [24]. Hou and his colleagues [25] have shown that uniformly dispersing multi-wall carbon nanotubes (MWCNTs) as reinforcement in a Mg-Zn matrix has improved the mechanical properties and thermal conductivity.

However, so far, research on the reinforcement function of CNTs for Mg-based composites has mainly focused on their mechanical properties (Table 1). Nonetheless, these reinforcing materials pose biosafety issues and may improve the antibacterial action of Mg-based composites for orthopedic applications. According to a review of the scientific literature, no attempts have been made to investigate Mg−2.5Zn−0.5Zr/CNTs nanobiocomposites. The aim of this research is to make nanobiocomposite Mg−2.5Zn−0.5Zr/CNTs with a mechanical alloying method and SPM process along with SPS, and investigate the feasibility of improved mechanical, corrosion and antibacterial performance for biomedical devices.

**Table 1.** A summary of the mechanical reinforcement performance of CNTs in Mg-based composites.

| Samples | Processing Route | Elongation(%) | Ultimate Compressive Strength, MPa | Hardness HV | Years | Ref. |
|---|---|---|---|---|---|---|
| MZ−3Zn | SPM + HTE | 12.1 ± 1.3 | 289.6 ± 13 | 66 ± 2 | 2022 | [26] |
| MZ−3Zn−0.2fCNT | SPM + HTE | 13.6 ± 1.5 | 368.2 ± 12 | 70 ± 2 | 2022 | [26] |
| MZ−3Zn−0.4fCNT | SPM + HTE | 15.7 ± 1.5 | 390 ± 15 | 74 ± 2.5 | 2022 | [26] |
| MZ−3Zn−0.8fCNT | SPM + HTE | 11.9 ± 1.3 | 320.2 ± 14 | 76 ± 3 | 2022 | [26] |
| AZ61 | PM | - | 135.7 | - | 2020 | [27] |
| AZ61−0.1CNT | PM | - | 127.2 | - | 2020 | [27] |
| AZ61−0.2CNT | PM | - | 146.8 | - | 2020 | [27] |
| AZ61−0.5CNT | PM | - | 168.4 | - | 2020 | [27] |
| AZ91 | PM | - | 141.2 | - | 2020 | [27] |
| AZ91−0.1CNT | PM | - | 132.3 | - | 2020 | [27] |
| AZ91−0.2CNT | PM | - | 145.6 | - | 2020 | [27] |
| AZ91−0.5CNT | PM | - | 153.5 | - | 2020 | [27] |
| AZ31 | PM + Extrusion | 14.5 ± 1.5 | 363 ± 3.5 | 58 ± 3.0 | 2015 | [28] |
| AZ31−0.3GNPs | PM + Extrusion | 21.7 ± 2.8 | 397 ± 5.3 | 71 ± 2.1 | 2015 | [28] |
| AZ31−0.3CNTs | PM + Extrusion | 13.3 ± 3.0 | 457 ± 6.0 | 78 ± 2.8 | 2015 | [28] |
| Mg−6Al−0.5CNT | MBM + CP + HTE | - | ~160 | ~40 | 2014 | [29] |
| Mg−6Al−1CNT | MBM + CP + HTE | - | ~140 | ~36 | 2014 | [29] |
| Mg−6Al−2CNT | MBM + CP + HTE | - | ~105 | ~34 | 2014 | [29] |
| Mg−6Al−4CNT | MBM + CP + HTE | - | ~75 | ~28 | 2014 | [29] |
| Mg−1Al | SPM + VS + HTE | 6.9 ± 0.5 | 377 ± 8 | 50 ± 4 | 2014 | [30] |
| Mg−1Al−0.6GNTs | SPM + VS + HTE | 4.0 ± 0.6 | 407 ± 3 | 63 ± 2 | 2014 | [30] |
| Mg−1Al−0.6CNTs | SPM + VS + HTE | 10 ± 0.3 | 425 ± 5 | 61 ± 5 | 2014 | [30] |
| Mg−1Al−0.6(1:5) (CNTs + GNPs) | SPM + VS + HTE | 16 ± 0.5 | 397 ± 3 | 56 ± 3 | 2014 | [30] |
| AZ81 | DMD + HTE | 7.9 | 487 ± 14 | 119 ± 2 | 2013 | [31] |
| AZ81−1.5CNTs | DMD + HTE | 12.9 | 488 ± 13 | 114 ± 8 | 2013 | [31] |
| ZK60A | DMD | 6.6 ± 0.6 | 522 ± 11 | 138 ± 7 | 2011 | [22] |
| ZK60A−1.0CNTs | DMD | 15.0 ± 0.7 | 547 ± 3 | 114 ± 6 | 2011 | [22] |
| Mg | PM-HTE | - | 239 ± 15 | 40 ± 2 | 2011 | [32] |
| Mg−0.5Al−0.18CNT | PM-HTE | - | 357 ± 13 | 50 ± 4 | 2011 | [32] |
| Mg−1Al−0.18CNT | PM-HTE | - | 421 ± 15 | 58 ± 3 | 2011 | [32] |
| Mg−1.5Al−0.18CNT | PM-HTE | - | 421 ± 11 | 60 ± 4 | 2011 | [32] |

HTE: hot extrusion, MBM: mechanical ball milling, PM: powder metallurgy, CP: cold pressing, VS: vacuum sintering, DMD: disintegrated melt deposition.

## 2. Materials and Methods

### 2.1. Raw Materials

Pure Mg (99.5%, <10 µm), Zn (99.9%, <3 µm), and Zr (99.9%, <3 µm) powders were obtained from Sigma Aldrich, USA. Multi-walled CNTs (diameter = 5–30 nm, length= 5–10 µm and purity = 95%) were provided by Platonic Nanotech Pvt. The SBF solution was purchased from ESPADANA Co., Iran. Fabrication of the base alloy with mechanical alloying method; Mg−2.5Zn−0.5Zr powders were prepared in a planetary ball mill. The enclosed

powders were placed in sealed 120 mL steel containers rotating at 300 rpm, and a mixture of balls (Ø = 10 mm balls with 4.07 g mass and Ø = 20 mm balls with 32.65 g mass) were placed in the containers (the ratio of balls to powder was about 20:1). Moreover, in order to prevent the oxidation of the raw materials, the air was taken out of the chambers and replaced by a neutral gas, argon (<3 ppm oxygen). In order to make Mg−3Zn−1Ca−0.5Zr/xCNTs (x = 0, 0.3, 0.6, 0.9) nanobiocomposites, the SPM method was used. For this purpose, CNTs and ethanol were put in the ultrasonic device to break the molecular bonds and prevent the van der Waals bonds between the CNTs. Then, Mg alloy was added to the solution and mixed for 2 h at a speed of 600 rpm at a temperature of 40 °C. Finally, it was placed in an oven with controlled atmospheric conditions for 1 day to dry. Composite powders with CNT values of 0, 0.3, 0.6 and 0.9 were labeled as MZ, MC1, MC2, and MC3, respectively. The composite powders were sintered in an SPS chamber at a temperature of 570 °C and a pressure of 40 MPa for 10 min. A schematic of the sample formation steps is represented in Figure 1.

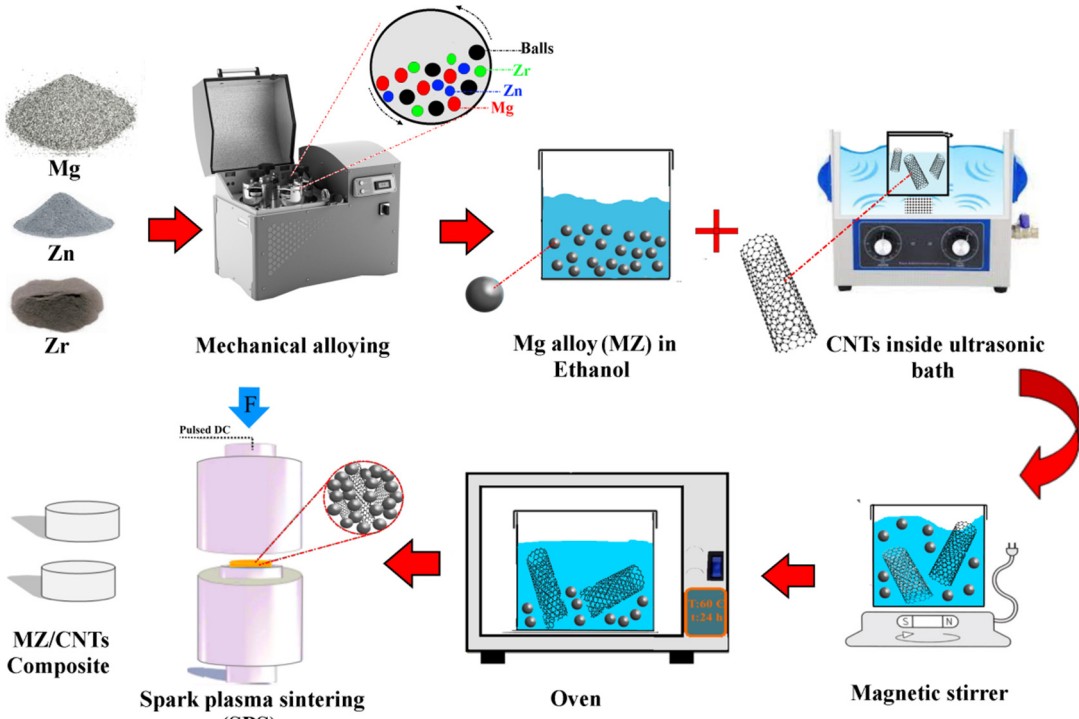

**Figure 1.** Illustration of the process of MZ/ CNTs composite preparation using SPM and SPS methods.

### 2.2. Microstructure and Mechanical Properties

The structure and morphology of the raw materials were investigated with optical microscopy (Olympus BX53M), scanning electron microscopy (SEM, QUANTAFEG250, FEI, USA) and transmission electron microscopy (TEM, HT7700 Hitachi, Japan) images. Energy dispersive X-ray spectroscopy (EDX, JSM−5910LV, JOEL Ltd., Japan) analysis was performed to complete the structural studies. Phase changes were studied with an X-ray diffractometer (XRD, D8 Advance, Brucker, Karlsruhe, Germany). Furthermore, Raman spectroscopy according to the literature [7] was used to determine the functional groups of CNTs.

Contact angle tests on the surfaces of the composites were performed using a static drop method via a video contact angle instrument (Dataphysics OCA 15) in air and at room temperature with a droplet size of 10 mL to measure the wettability. The results of the three samples were utilized to calculate the average contact angle for each composite. A SANTAM model device (STM50) was used to measure a cylindrical composite (diameter = 10, height = 15 mm) according to the ASTM-E9 standard at a velocity of 2 mm

per minute and a load of 10 kilonewtons. The test was repeated three times and the mean of the three data was reported in the final analysis. A Vickers microhardness test (LECO M-400) with a peak load of 300 g was also applied to study the microhardness of the composites. Five different locations were analyzed from each sample to obtain the results.

### 2.3. Degradation Behavior

To measure the degradation rate of composites in accordance with the ASTM-G31-72 standard [33], cylindrical specimens with a diameter of 10 mm and a height of 10 mm were placed in an incubator at pH 7.4 and a temperature of 37 °C, and they were soaked inside the incubator to prevent evaporation for 2 weeks. The pH change in the immersion test was recorded and evaluated at 12 h intervals. To assess hydrogen release, the released hydrogen bubbles were collected in a funnel and the volume change of the Kokubo SBF (A Kokubo SBF is a solution with ionic concentrations similar to human blood plasma (see Table 2)) was measured by a standard graduated burette attached to the funnel. A schematic of the experimental setup is presented in Figure 2. After washing with water and acetone to remove corrosion products from the surface of the sample, the sample was weighed. The formula $W/DAT \times 87.6 = CR$ was used to calculate the corrosion rate [34]. In this formula CR equals the typical corrosion rate in mm/year, as well as W, D, A and T are weight lost in milligrams, density, sample surface area exposed to the corrosion solution ($cm^2$), and exposure time, respectively.

**Table 2.** Chemical composition of the Kokubo simulated body fluid (SBF) compared to the human blood plasma.

| Solution | Ion Concentration (mmol/L) | | | | | | | |
|---|---|---|---|---|---|---|---|---|
| | $Na^+$ | $K^+$ | $Ca^{2+}$ | $Mg^{2+}$ | $HCO_3^-$ | $Cl^-$ | $HPO_4^{2-}$ | $SO_4^{2-}$ |
| **Plasma** | 142.0 | 5.0 | 2.5 | 1.5 | 27.0 | 103.0 | 1.0 | 0.5 |
| **Kokubo (c-SBF)** | 142.0 | 5.0 | 2.5 | 1.5 | 4.2 | 147.8 | 1.0 | 0.5 |

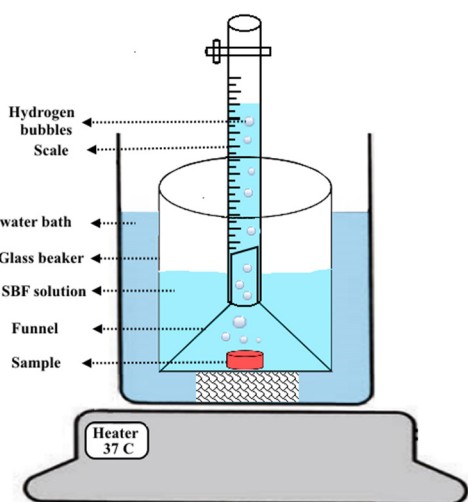

**Figure 2.** Schematic view of the hydrogen evolution measurement setup.

### 2.4. Antibacterial Evaluation

Gram-positive *S. aureus* and Gram-negative *E. coli* were used to assess the antibacterial performance of composites by a disc diffusion process.

To do this, a sterile swab was inserted into it, the microbial suspension was washed (squeezing the swabs to the side of the tube) and the culture medium was in the form of cultivation. The samples were placed in an incubator at 37 °C for 1 day. Note that

gentamicin discs are employed as antibiotics. If the samples have antibacterial properties, this can be recognized by their surrounding zones of inhibition (IA).

### 2.5. Biocompatibility Assessment

All samples were sterilized by ultraviolet (UV) irradiation for at least 2 h prior to cell testing. All samples were assessed for cell viability using the MTT (3-(4,5-dimethylthiazol-2-yl)-2,5-diphenyltetrazolium bromide) approach (n = 3), as shown in [24].

### 2.6. Statistical Analysis

Test outcomes are expressed as mean $\pm$ standard error (SE) and examined for significant data at *p*-values of 0.05 (*), 0.01 (**), and 0.001 (***) using Sigmaplot software.

## 3. Results and Discussions

### 3.1. Microstructure

Figure 3—SEM images show pure elemental Mg (a,b), Zn (c,d), and Zr (e,f) without mechanical or chemical stress. Pictures (g,h) show the alloy obtained from the grinding process after an optimal time of 25 h. Powder particles are constantly expanding, galling, breaking, and remelting during high energy milling. Decreases in particle size with increasing milling time have been observed. The result is an alloy powder (MZ) with a new surface created by particle-to-particle fusion and penetration.

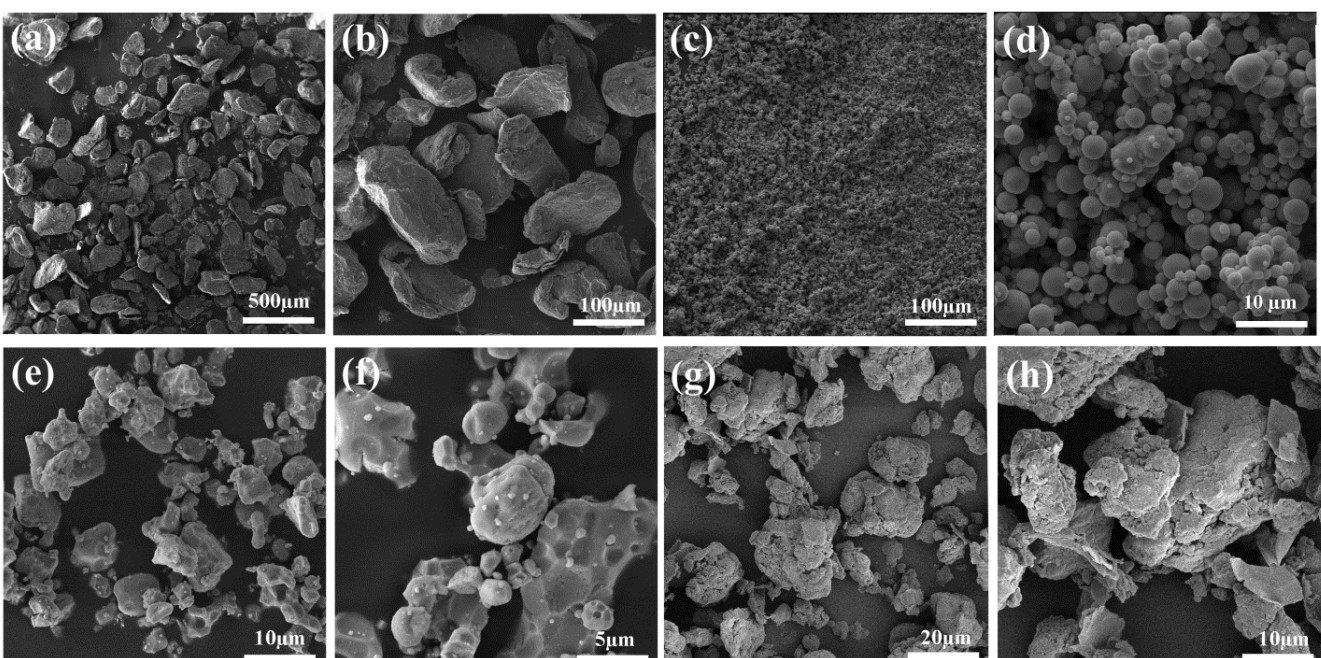

**Figure 3.** SEM micrographs of (**a**,**b**) Mg, (**c**,**d**) Zn, (**e**,**f**) Zr and (**g**,**h**) MZ powders.

Electron microscope image and EDX measurements of as-prepared samples are shown in Figure 4. According to the obtained results, Mg, Zn, Zr and O elements for the MZ alloy matrix (Figure 4a) and Mg, Zn, Zr, O and C elements for different amounts of CNT in the MZ/CNTs composites (Figure 4b–d) were identified. The pictures show the uniform distribution of the alloy elements by the mechanical alloying method, and the almost uniform distribution of the carbon reinforcement in the background phase by the SPM technique (Figure 4d).

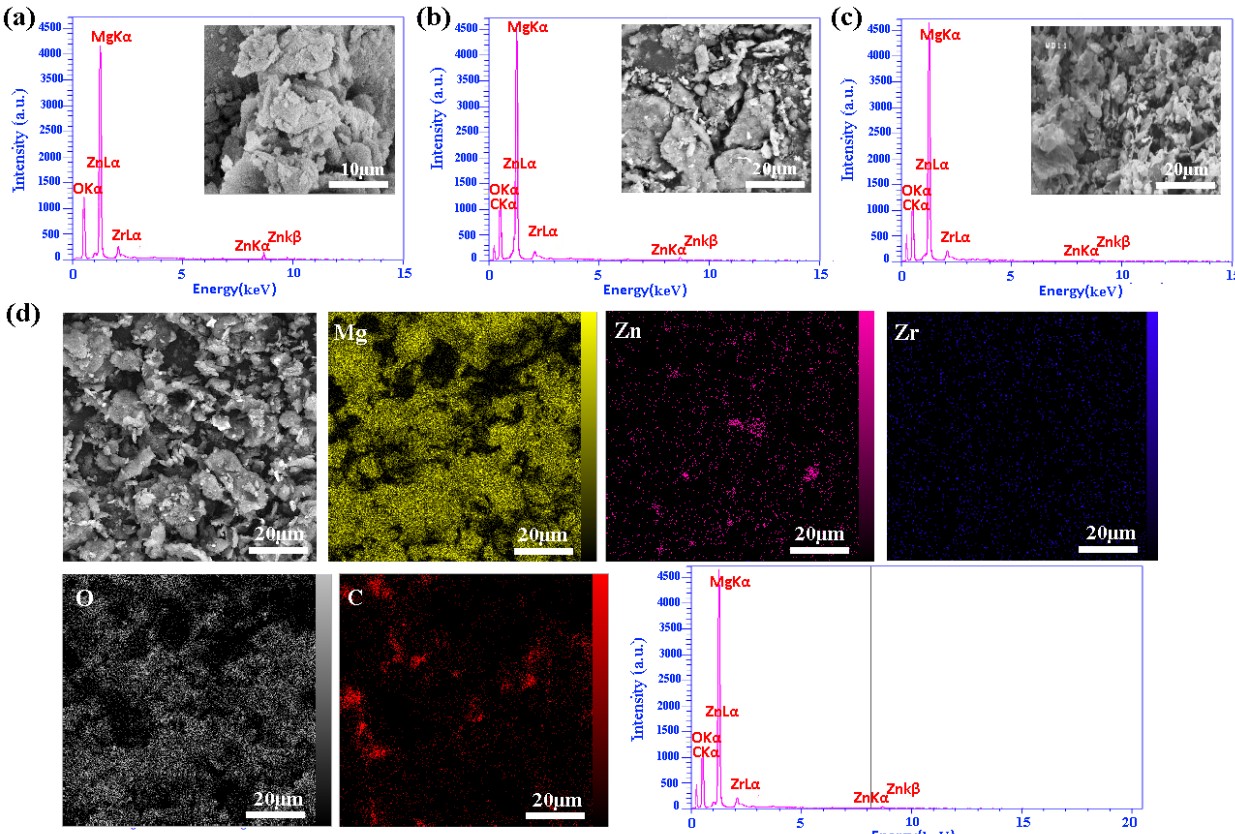

**Figure 4.** SEM images and EDX spectra of (**a**) MZ (**b**) MC1 (**c**) MC3 and (**d**) map images of MC2 powder composite.

Figure 5a shows the TEM micrographs of CNTs after the dispersion method (in ethanol solution) used in this study. As can be seen, individual CNTs had diameter sizes in the range of 50–60 nm. On the other hand, MC2 composite powders displayed a nanotube shape with a thickness of 50 nm and a homogeneous distribution of matrix particles (Mg alloy) inside the nanotube with high adhesion, as shown in Figure 5b–d. Considering that almost all Mg particles are adsorbed on the CNT surface and very few Mg particles are outside the CNT support, a strong MZ/CNT interaction was observed. Raman spectroscopy results for MWCNTs (Figure 5e) confirm the presence of D band (1335 cm$^{-1}$), G band (1567 cm$^{-1}$) and D' band, new second order D band (2686 cm$^{-1}$) which is a typical characteristic for multi-walled CNTs [35].

Figure 6—XRD analysis was used to identify elements and classify phases better. Only one set of clearly defined Mg-α peaks was found in pure Mg (standard card No. 190239) and there was no peak for the second phase. Clearly, the peaks detected at 2Θ are equivalent to 32.2, 34.4, 36.6, 47.5, 57.3, 63.0, 67.3, 68.6, 70.0 and 72.4° [36]. In addition, the XRD pattern shows peaks for Zn and Zr metals in the pure sample, while no peaks were found for the second phase [15,37,38]. The crystalline nature of the prepared CNTs sample can be confirmed by detecting the peak in the XRD spectrum [39]. In the XRD result related to the MZ/CNTs composite, the (002) peak of CNT is not clearly observed, which can be caused by the low amount of additives and also the way of preparation in the solution, which is associated with breaking van der Waals bonds and preventing local agglomeration. According to the XRD spectrum, only peaks related to α-Mg, MgZn$_2$, and Mg$_7$Zn$_3$ phases can be identified.

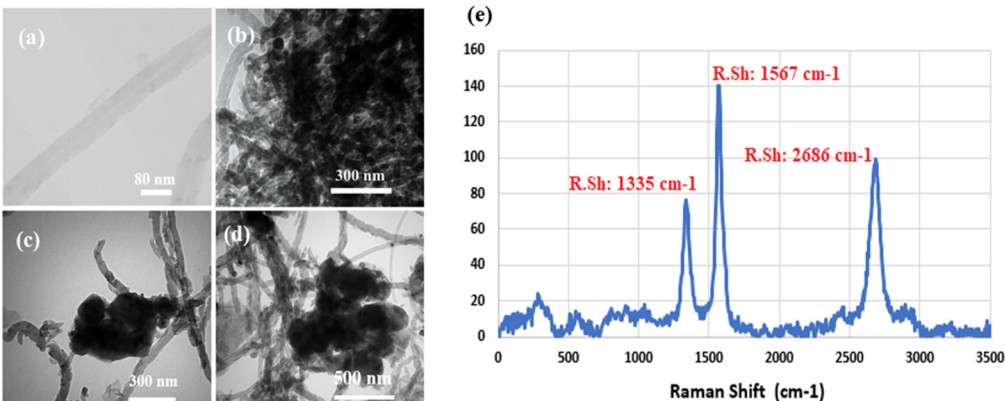

**Figure 5.** TEM micrographs of (**a**) CNTs, (**b–d**) MZ/CNTs powders and (**e**) Raman spectra of pure CNTs.

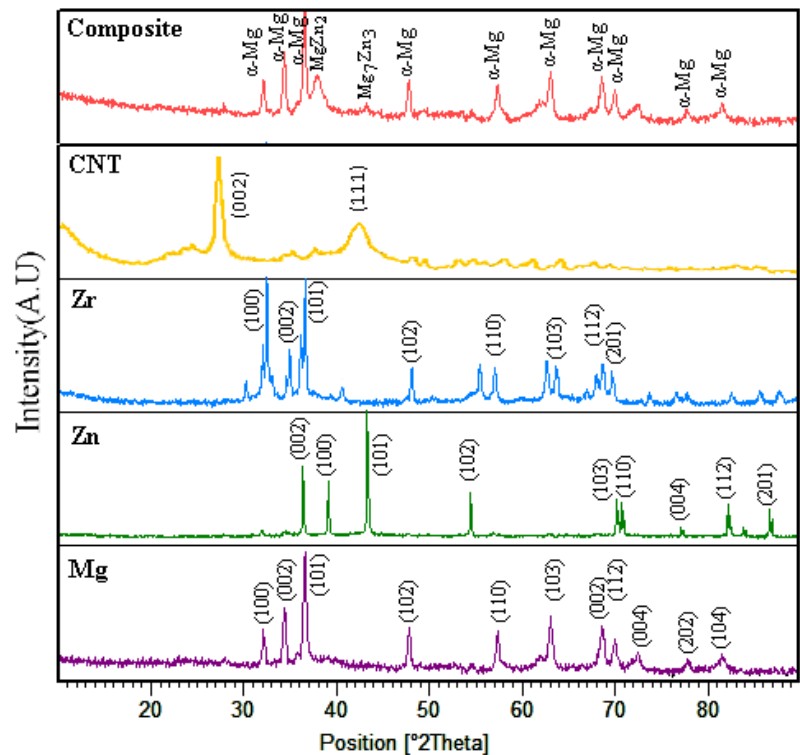

**Figure 6.** XRD patterns of pure Mg, Zn, Zr and MZ/CNTs composite.

Figure 7—shows the contact angles of a liquid drop with the bulk surface for MZ 120 and for MC1, MC3, MC2 are 97, 85, 76 degrees, respectively. The presence of CNTs affects the surface properties of MZ/CNTs composites. Adding 0.3 wt% causes a sharp increase in contact surface. The reason can be ascribed to the formation of functional groups such as carboxyl, carbonyl and hydroxyl, which leads to the hydrophilicity of the sample surface. This phenomenon is directly related to the improvement of biocompatibility, as much as the hydrophilicity of Mg-based composites improves, the biocompatibility increases [40,41]. This may significantly contribute to medical success, as improved wettability of the implantable composite surface improves its ability to adhere to biological materials. As examined in previous studies, increasing the hydrophilicity of the bone implant surface is advantageous for the absorption of nutrients and bioactive factors, thereby promoting bone healing during in vivo implantation [42].

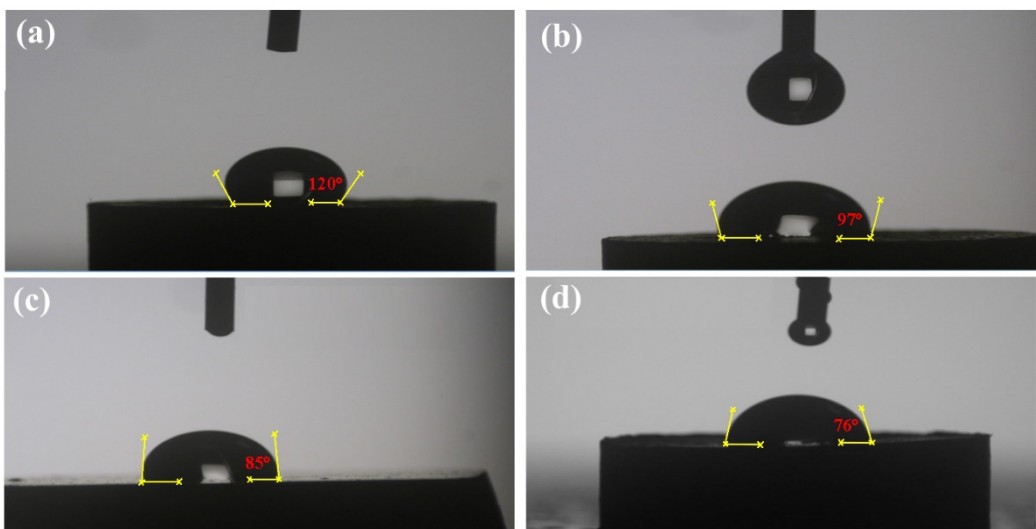

**Figure 7.** Contact angles of (**a**) MZ, (**b**) MC1, (**c**) MC2 and (**d**) MC3 composites after SPS process.

*3.2. Mechanical Properties*

Figure 8a displays the microhardness of MZ/CNTs composites, which increases by increasing the amount of CNT from 0, 0.3, 0.6, and 0.9 wt% to 58, 67, 78, and 80 HV, respectively.

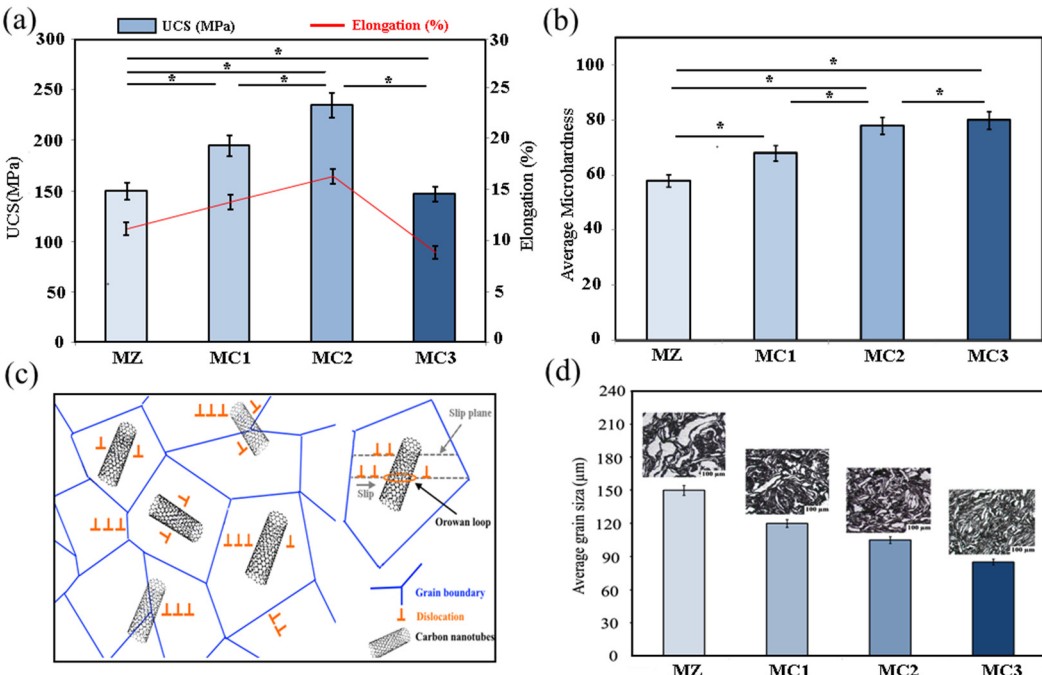

**Figure 8.** (**a**) Microhardness value, (**b**) ultimate compressive strength (UCS) value of MZ, MC1, MC2 and MC3 composites, (**c**) schematic showing the Orowan looping mechanism after dislocations gliding over and bypassing the CNTs, (**d**) OM image and grain size calculations of MZ, MC1, MC2 and MC3 composites.

Another important criterion for a bone replacement implantable device is compressive strength [43]. Figure 8b shows the results of the compressive strength tests by plotting two parameters of UCS and elongation of MZ/CNTs composites. The results show that composites containing 0.3 and 0.6 wt% CNTs have higher UCS values and higher elongations at break than the base matrix, with the highest stiffness associated with MC2 composites.

On the other hand, when the amount of CNTs was increased, a decrease in mechanical properties was observed. This may be attributed to the phenomena of aggregation and van der Waals binding. Among the possible mechanisms that play a role in the increase in compressive strength, the roles of CNTs as crack bridging and crack deflection inhibitors have been mentioned [44]. In other words, there is an effective load transfer from Mg to the reinforcement due to the higher elastic modulus of CNTs compared to the Mg matrix and failure strain [45]. Another dominant mechanism is the Orowan mechanism. A scheme presenting the Orowan ring mechanism after dislocations slide on and around the CNTs is shown in Figure 8c. The contribution of different strengthening mechanisms depends on the CNT volume fraction, CNT surface bonding, matrix and base alloy type, and the CNT aspect ratio [44,46,47]

Figure 8d shows the plots along with optical micrographs of MZ/CNTs. A significant decline in particle size was observed with the addition of CNT reinforcement. That is, the improvement in the mechanical properties of nanocomposites can be attributed to grain refinement due to the presence of CNTs [48].

*3.3. Assessment of Degradation Behavior*

Figure 9—shows the surface morphology of Mg alloys and MZ/CNTs composites after being immersed in SBF for 7 days. In Figure 8a, many deep cracks and pitting corrosion are observed, indicating severe corrosion damage to the Mg alloy. However, the MZ/CNT composite with a small amount of CNTs has fewer cracks and smaller pits (Figure 8b,c). Hard corrosion and deep cracks were observed with increasing CNT content (Figure 9d).

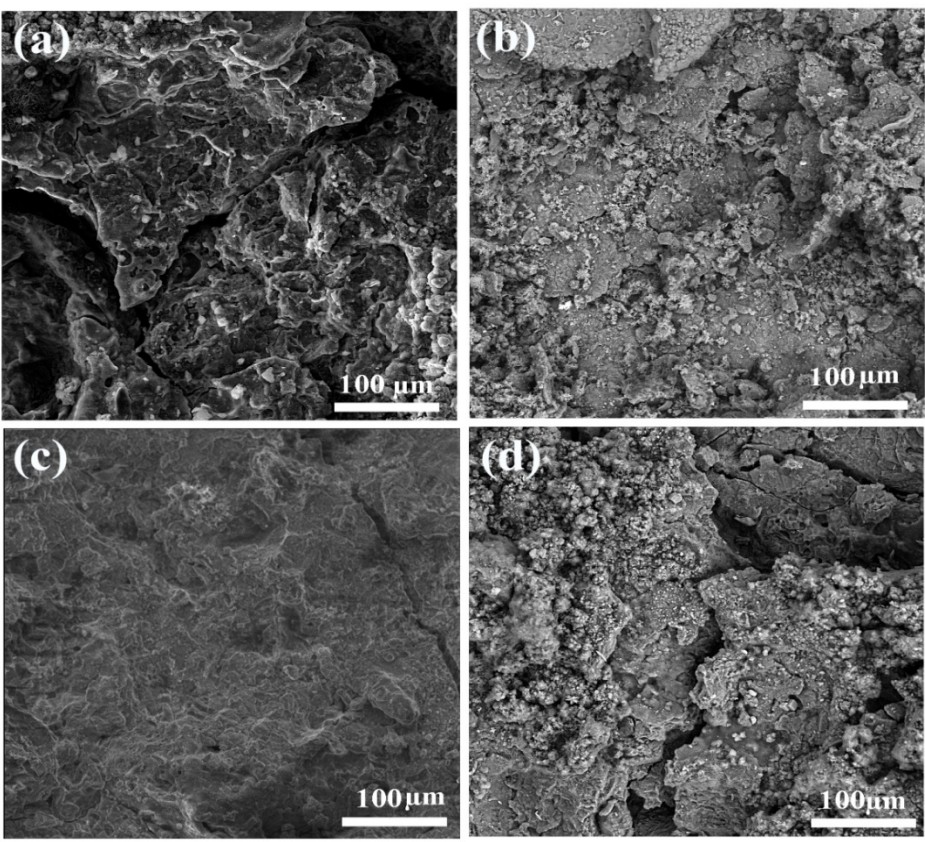

**Figure 9.** Surface morphology of the (**a**) MZ, (**b**) MC1, (**c**) MC2, (**d**) MC3 composites after being immersed in SBF for 7 days.

Carbon nanomaterials (CNMs) such as CNTs seem to have a significant impact on heterogeneous nucleation, causing grain refinement of the Mg-α based phase and enhancing the corrosion resistance of the alloys [7]. Therefore, the decomposition rate of the Mg-α

based phase in composites with a low CNT share is slowed down and less $Mg(OH)_2$ and $H_2$ are produced. On the other hand, CNTs have a bridging function and their filamentous appearance can prevent or retard delamination of the surface oxide layer of composites, which has a positive effect on corrosion resistance [49]. Furthermore, apatite deposits can form dense layers on Mg composites with the oxygen-rich groups and the available sites of CNTs, and prevent the penetration of the corrosive solution into the sample [50].

Figure 10a shows the XRD pattern of MC2 after seven days of immersion in the SBF solution. It is clear that the corrosion products $Mg(OH)^2$ and HA occur because of the presence of inorganic ions including $H_2PO^{4-}$, $Ca^{2+}$ and $Cl^-$ in the SBF solution [51].

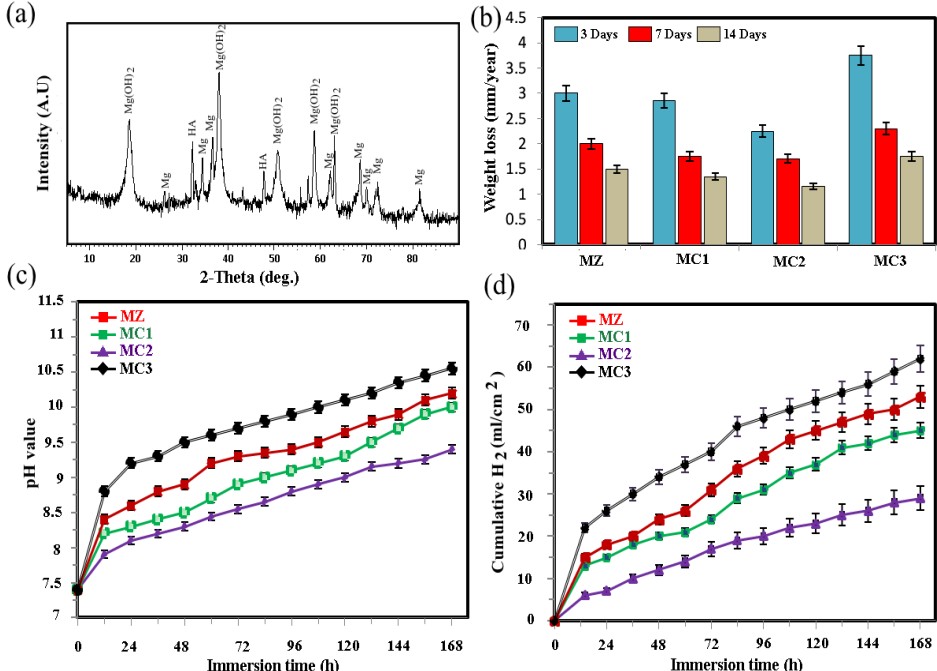

**Figure 10.** (**a**) X-ray diffraction patterns of MC2 after soaking in SBF for 7 days, (**b**) weight loss corrosion rate of MZ/CNTs nanocomposites in Kokubo solution for the durations of 3, 7 and 14 days, (**c**) pH value, (**d**) H2 evolution of MZ, MC1, MC2 and MC3 composites.

Figure 10b shows a plot of the weight loss of the composite after 3, 7 and 14 days of immersion. As can be seen, the corrosion rate of the sample increased during the first three days. This is likely as a result of the surface area of the sample being directly exposed to the solution for the first few days, causing an exothermic reaction and accelerating the corrosion rate. Moreover, due to the comparatively high chloride concentration of the Kokubo solution, this phenomenon may also occur in the first few days. By extending the duration of the immersion test to seven days, the degradation rate of 0.3 and 0.6 wt% CNT nanocomposites increased at lower exposure gradients compared to the first three days of exposure. However, nanocomposites comprising 0.6 wt% CNTs showed lower corrosion rates at this exposure time. A 14-day immersion period decreased the corrosion rate of all nanocomposites. This may be due to the creation of a protective layer of corrosion products on the surface of the nanocomposites after long-term exposure. This layer can diminish the corrosion rate of nanocomposites through impeding direct contact of the nanocomposites to the solution [43]. The pH of the solution was also monitored during the immersion test to detect the release of degradation products (Figure 10c). A gradual increase in pH was observed for all samples. The increased alkalinity of the SBF solution can be ascribed to the diffusion of $OH^-$ ions [52]. The pH value of the electrolyte increased rapidly with the immersion time at first, and then gradually and finally steadied at constant values (Figure 10c). The mechanism of the reactions is as follows:

I.     Cathodic decomposition of water,

$$H_2O + 2e^- \rightarrow 2OH^-_{(aq)} \tag{1}$$

II.    Anodic dissolution of α-Mg

$$Mg_{(s)} \rightarrow Mg^{2+}_{(eq)} + 2e^- \tag{2}$$

III.   Creation of a hydroxide layer

$$Mg^{2+}_{(eq)} + 2OH^-_{(aq)} \rightarrow Mg(OH)_{2(S)} \tag{3}$$

The release of hydrogen bubbles immediately after placing the sample in the solution, which indicates the beginning of the reaction between the composite and the solution, was also evaluated. Figure 10d shows that the MC2 sample significantly reduced $H_2$ released from $62 \pm 3$ to $29 \pm 3$ mL/cm$^2$ compared to the Mg alloy matrix.

*3.4. Biological Properties*

3.4.1. Antibacterial Evaluation

In order to evaluate the potential antibacterial properties, all the samples were exposed to Gram-negative and Gram-positive bacteria, that is, *E. coli* and *S. aureus* models. The zone of inhibition (in millimeters (mm)) was then measured for different samples.

Figure 11a shows that the bacterial growth stopped around the MZ/CNTs nanocomposites containing 0.3–0.9 wt% CNTs, while bacterial proliferation was detected around the MZ matrix. In addition, the images show the formation of a wider zone of inhibition around the composite MC3 (3.3 mm) on the agar plate compared to the composite MC1 (2.05 mm), which contains fewer CNTs. Therefore, it can be concluded that the antibacterial properties of nanocomposite samples are related to their CNTs content, that is, increasing the content of CNTs in the nanocomposite is associated with larger inhibition zones. While the inhibition zones of E. coli and S. aureus were in the ranges of 0.24–3.1 mm and 0.33–3.3 mm, respectively. The antibacterial activity of CNMs depends on their composition, target microorganisms, surface modification, and reaction environment. The antibacterial mechanism of CNMs is based on attacking the membrane/wall of microbial cells, thereby damaging cellular structures and the physical mechanisms associated with the biological separation of microbial cells from the environment [53,54]. It also creates oxidative stress conditions through the generation of toxic substances, such as reactive oxygen species (ROS), and chemical antimicrobial effects that depend on the interaction of micro-organisms with CNMs. CNM-microbe interactions facilitate electron transfer. ROS-independent oxidative stress is promoted by the removal of electrons from the microbial surface leading to biological death [55].

According to some research, CNTs can exhibit full antibacterial activity. In fact, size plays an important role in microbial inactivation. Indeed, the surface-to-volume ratio of CNMs increases with decreasing size, resulting in a stronger attachment to the microbial membrane or cell wall and more effective microbial tasks. These mechanisms are reliant on the ability of CNTs to bind to micro-organisms and disrupt their cell membranes, morphology, and metabolic processes [55]. The bacteriostatic properties of CNTs have been shown to result from their ability to damage microbial cell membranes and cause bacterial cell death upon direct contact. Upon incubation with CNTs, micro-organisms exhibit morphological changes of cell integrity disruption. Furthermore, a five-fold improvement in plasmid DNA, release of cytoplasmic material, and a two-fold increase in RNA were demonstrated after exposure to small CNTs [56].

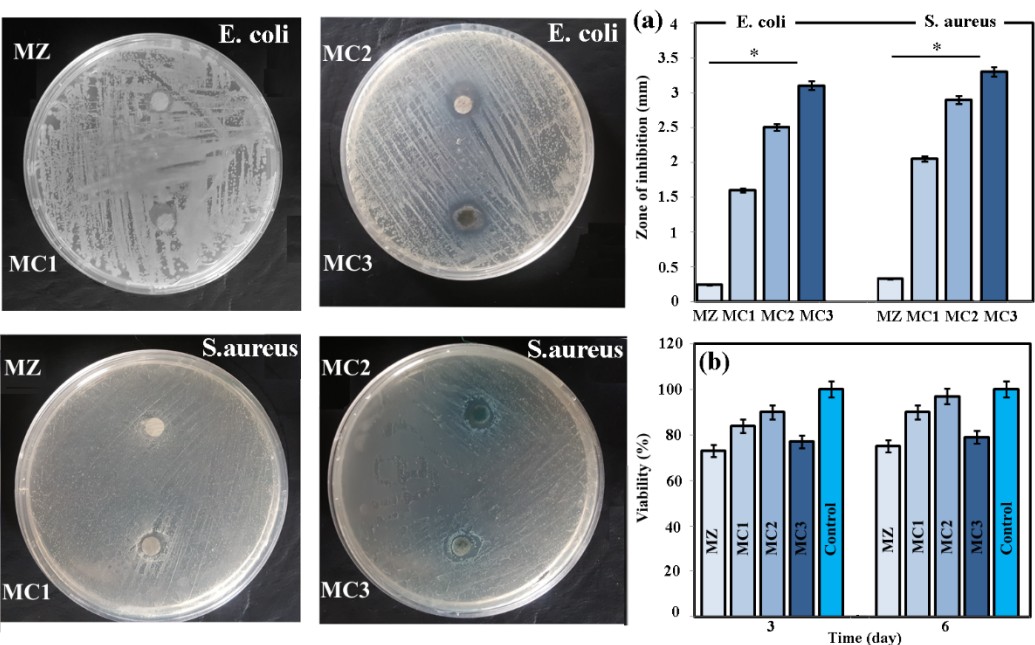

**Figure 11.** (**a**) Antibacterial activities by the disk diffusion test against both gram-positive (*S. aureus*) and gram-negative (*E. coli*) and (**b**) cell viability of MG63 cells cultured for various times on the *MZ, MC1, MC2* and *MC3* biocomposites.

### 3.4.2. Cellular Compatibility

Figure 11b shows the results of an MTT assay of MG63 cells cultured on nanocomposites for three and seven days. The number of viable cells is proportional to the amount absorbed. In particular, increasing the culture time slowly increases the number of viable cells in the MZ/CNT nanocomposites. MC1 and MC2 composites are effective in stimulating cell viability compared to the controls. The addition of large amounts of CNTs up to 0.9 wt% resulted in a strong inhibitory effect during long-term incubation. After three days of incubation, the cell viabilities of MZ, MC1, MC2, and MC3 were 73, 84, 90, and 77%, respectively, which are higher than 75% for composites containing CNTs, indicating a good cell compatibility [57]. Increasing the CNT content may lead to greater destruction by galvanic corrosion and increased toxicity [58–60].

### 4. Conclusions

The microstructure, mechanical characteristics, degradation and antibacterial activity of Mg−2.5Zn−0.5Zr/xCNTs nanocomposites (x = 0, 0.3, 0.6 and 0.9) made with mechanical alloying processes and SPM with SPS were evaluated. The increase in compressive strength of MZ/CNT composites was evident compared to the base alloy without CNTs. The addition of low concentrations of CNTs to Mg-based composites reduced the degradation strength in the SBF environment by almost half. According to the cytotoxicity studies, composites with low CNT concentrations (0.6 wt%) showed good biocompatibility. All MZ/CNT composites exhibited excellent antibacterial properties against *E. coli* and *S. aureus*. The antimicrobial activity of the composites was shown to increase with increasing amounts of CNTs. Therefore, Mg−2.5−0.5Zr/0.6CNT composites with excellent performance in treating bone infections can be a substitute for implantable devices in biomedical applications.

**Author Contributions:** Writing—original draft preparation, formal analysis, J.Z.; Conceptualization and methodology, formal analysis, writing—review and editing, M.H., A.S. and Z.H.; writing—review and editing, funding acquisition, M.S.B. All authors have read and agreed to the published version of the manuscript.

**Funding:** This work was supported by an innovation grant of the TUIASI, project number MedTech_ 8 /2022.

**Institutional Review Board Statement:** Not applicable.

**Informed Consent Statement:** Not applicable.

**Data Availability Statement:** All data provided in the present manuscript are available to whom it may concern.

**Conflicts of Interest:** The authors declare no conflict of interest.

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
