# Peer review of "Carbon Nanotube (CNT) Encapsulated Magnesium-Based Nanocomposites to Improve Mechanical, Degradation and Antibacterial Performances for Biomedical Device Applications"

_coatings, doi:10.3390/coatings12101589_

Round 1

Reviewer 1 Report

The manuscript presents the results form a study on the effect of carbon nanotubes on  the mechanical, degradation and biological performance of magnesium-based nanocomposites for implant applications. The text is well organized, the methodology is well chosen, the presentation of the results is clear and convincing, the conclusions are supported by the presented data, the references are appropriately cited.

In my opinion the study and the obtained results will be of interest for the Journal readers as it presents a step towards improving the  properties of a material envisaged for bone implants. The research is actual, modern means for assessment of the properties on the studied materials are used, recently published references are cited to support the authors observations and conclusions.

I recommend the paper to be published after minor revision. The corrections which I recommend below are connected mainly to the text and the discussion, not to the quality of the obtained results. The main recommendation is to compare the obtained results with published nanocomposites based on Mg and CNT.

Please, find the detailed comments in the attached file.

Author Response

First, all authors appreciate the reviewers' valuable and practical comments for improving our article. We have revised our article according to the comments.

In this letter, the answers to all comments have been organized one by one. The comments have been marked as Q. and the answers as A. The answers to your comments were also added in the manuscript's main text as highlighted.

Reviewer 2 Report

This paper is revealing nanocomposite synthesis and the potential use would be for implanting media for organ growth.

Although the authors report lengthy data characterizing nanocomposite properties. However no experiment was done using actual mammalian cells that are supposed to grow as an organ. We need to ask them to revise the manuscript for adding cell implanting experiment data.

Some points to improve the paper are as follows.

Zn is the second most abundant element in the body after Fe.

Actually, the second most abundant element in the body is carbon after oxygen. Zn is about 0.003 % of our body and 0.004 % of Fe. 

TEM data explanation, Fig. 4 (a-b) shows in page 5-6 is not clear, some particulates are observed in (a) and CNT seems to be folded each other although the authors say that there are no CNT agglomerates, and Fig. 4 (c-d) shows that Mg particles are aggregated though the authors claim that they are well distributed. The authors need to show a wide area view of TEM for overall comparison. 

120 ml steel containers rotating at 300…”  line 104, page 3,  ml should be mL" . 

size of 10 ml to measure the wettability.  line 134, page 4, ml should be mL" .

Textural vagueness is observed such as  A less visible decrease in particle size with increasing milling time has been observed.  This can be corrected as Decrease in particle size with increasing milling time has been observed.

In figure caption 2, Figure 2. SEM micrographs of (a,b) Mg, (c,d) Zn, (e,f) Zr and MZ powders. (g,h) is omitted.

EDX X-ray diffraction spectroscopy shows the..  in 183, page 5, EDX X-ray diffraction spectroscopy must be corrected as Energy-dispersive x-ray spectroscopy.

Author Response

(The authors gave the same response as above.)

Reviewer 3 Report

In this study, biocomposite Mg-2.5Zn-0.5Zr/CNTs were prepared and mechanical, corrosion and biological performance were assessed. Some comments were given in order to improve the manuscript:

Materials and Methods

1. Do you examine the chemical compositions of MZ, MC1, MC2 and MC3 by EDX or XPS?

2. p.4 of 14, 2nd paragraph, “A SANTAM model device (STM20) was used ….”. Please check it is used to measure the compressive strength.

3. What is the composition of SBF used for the degradation test?

4. Could you describe in more details how to measure the hydrogen gas generated? Do you have the photo image/diagram to show it?

Results and discussions

1. The contact angle decreased as more carbon nanotubes were added into the Mg alloys. This means the surface of Mg alloys is more hydrophilic. Could you explain in little details that an increase of the hydrophilicity of an implant’s surface is beneficial for the implant?

2. In Fig. 8, did you also evaluate the change of mineral content, e.g. Ca and P after the Mg alloys were immersed in SBF for 7 d?

3. Fig. 9a showed the X–ray diffraction patterns of MZ/CNTs after soaking in SBF for 7 days. Which MZ/CNTs was it, MC1, MC2 or MC3?

4. In Fig 9c and d, please label which one is MZ, MC1, MC2 and MC3.

5. Which one, MZ, MC1, MC2 and MC3 has the best corrosion resistance?

Author Response

(The authors gave the same response as above.)

Round 2

Reviewer 2 Report

Revision is done following the comments  partly

Author Response

The paper was improved with the corrections made by the reviewer.
